# Analysis of the Nutritional Status in Homeless People in Poland Based on the Selected Biochemical Parameters

**DOI:** 10.3390/ijerph18052340

**Published:** 2021-02-27

**Authors:** Edyta Naszydłowska, Anna Cedro, Edyta Suliga, Dorota Kozieł, Kamila Sobaś, Anna Jegier, Stanisław Głuszek

**Affiliations:** 1Department of Nursing, Midwifery and Emergency Medicine, Institute of Health Sciences, Jan Kochanowski University, ul. Zeromskiego 5, 25-369 Kielce, Poland; edyta.naszydlowska@ujk.edu.pl (E.N.); dorota.koziel@wp.pl (D.K.); 2Department of Cardiovascular Disease Prevention and Pharmacology, Institute of Health Sciences, Jan Kochanowski University, ul. Zeromskiego 5, 25-369 Kielce, Poland; 3Department of Nutrition and Dietetics, Institute of Health Sciences, Jan Kochanowski University, ul. Zeromskiego 5, 25-369 Kielce, Poland; edyta.suliga@ujk.edu.pl (E.S.); ksobas@ujk.edu.pl (K.S.); 4Department of Sports Medicine, Chair of Social and Preventive Medicine, Medical University of Lodz, ul. Pomorska 251, 92-213 Lodz, Poland; anna.jegier@umed.lodz.pl; 5Department of Surgical Medicine with the Laboratory of Medical Genetics, Institute of Medical Sciences, Jan Kochanowski University, ul. Zeromskiego 5, 25-369 Kielce, Poland; sgluszek@wp.pl; 6Department of Clinic General Oncological and Endocrinological Surgery, Regional Hospital, Prosta 30, 25-371 Kielce, Poland

**Keywords:** body mass index (BMI), malnutrition, overweight, homeless people, nutrition indicators

## Abstract

The aim of the study was to assess the nutritional status of adult homeless people using both anthropometric and biochemical measurements. The analysis comprised anthropometric indicators, i.e., body mass index and waist circumference, and the following biomarkers: red blood cells, hemoglobin, hematocrit, mean corpuscular volume, mean corpuscular hemoglobin concentration, mean corpuscular hemoglobin, white blood cells, complete lymphocyte count, neutrophils-to-lymphocytes ratio, platelets-to-lymphocytes ratio, platelets-to-leukocytes ratio, C reactive protein level, serum iron concentration, serum albumin concentration, total serum protein, fasting lipids and blood glucose level. There were representative Polish homeless people enrolled (*n* = 580). The analysis of the conducted studies proved that there is a greater frequency of overweight and obesity than underweight in the target population. The major problem was abdominal obesity that was present statistically more frequently in women than men (*p* < 0.001). In the majority of cases, homeless people were found to have normal complete blood count parameters. In obese people, there were statistically significant both elevated and decreased hematocrit levels, a significant decrease in red blood cells, elevated serum glucose, triglycerides and total protein level (*p* < 0.05). The presence of abdominal obesity, elevated glucose concentration, low-density lipoprotein cholesterol, and triglycerides, and decreased high-density lipoprotein cholesterol in serum together with smoking increase the risk of cardiovascular disease.

## 1. Introduction

Polish homeless people are mainly men, in the predominant age group of 40 and 60, in some areas 51–60 years of age, living alone, mainly in agglomerations and large cities, mostly with basic vocational or lower education and usually professionally inactive and/or unemployed. Most of the Polish homeless people, about 60%, stay in various types of institutional facilities, and the remaining people stay in nonresidential places (stations, canals, chutes, plots and gazebos) or temporarily (not voluntarily and without registering) with friends or family.

The causes of homelessness are usually: individual (e.g., addictions, disability, mental disorders), social (e.g., family conflicts, relationship problems, domestic violence), institutional (leaving institutions), structural (poverty, unemployment, housing problems). The main source of income for the homeless is social benefits, including social assistance [1].

In Poland, in 2014, the Commune Model of Getting Out of Homelessness (GSWB) was developed, indicating activities in the field of homelessness at the level of prevention, intervention and integration [2]. The analysis of the present results of research relating to the health of the homeless shows that they are too general and, according to the authors, require an in-depth diagnosis also based on objective indicators.

Nutritional status is defined as health status resulting from usual food consumption, digestive processes, absorption, use of nutrients and the influence of pathological factors on all the mentioned above [3]. Human nutrition, especially the type and amount of consummated products, should be tailored to individual needs and prophylaxis of diet-related diseases [4,5]. Inappropriate nutrition may be the reason for the elevated risk of chronic diseases. The increasing prevalence of these diseases has a direct influence on healthcare and makes the problem interesting among both researchers and decision-makers at different levels of management [6]. Based on the statistical analyses, one can claim that inappropriate diet in Poland in 2016 resulted in nearly 16% loss in years in health (17% in man, 13% in women, respectively) and was the main cause of death, and in women, the reason of disability [7]. Worldwide public health problem of major concern is overweight and obesity, especially abdominal obesity, which is one of the main causes of metabolic syndrome [8,9,10,11]. Based on the available studies, one can claim that the compilation of overweight, obesity and smoking can lead to a significant increase in mortality and death risk, which are possibly preventable [12]. What is more, low socioeconomic status may be connected with inappropriate nutritional status and its after-effects [13,14]. One can suspect, however, that these would be malnutrition, diseases, weakness due to long-term lack of adequate amount and quality of food that are dominant matters in the homeless population not overweight and obesity [15].

The aim of the study was to measure anthropometric and biochemical indicators of nutritional status in adult homeless people temporarily staying in shelters and hostels in Poland. The analysis of the results of research to date relating to the health of the homeless shows that they are too general and, according to the authors, require an in-depth diagnosis also based on objective indicators. The diagnosis of the nutritional status at the national level is of key importance not only for emergency aid but also for the prevention of homelessness and social reintegration of the homeless.

## 2. Materials and Methods

### 2.1. Study Design and Sample Collection

The study enrolled a representative sample of 614 homeless individuals, among them 104 women aged 21–79, mean 49.0 ± 13.6 years and 510 men, aged 18–79, mean 53.7 ± 11.6 years, staying in shelters and hostels in Poland in years 2011–2012. From February to November 2011, blood was collected from the homeless, and the necessary measurements were made, and the collection and analysis of the results were carried out in 2012. Due to the specificity, the study included homeless people staying in institutions with the status of shelters and hostels. The list of institutions in Poland was downloaded from the website of the Ministry of Labor and Social Policy [16]. The facilities were selected by the method of two-tier random selection. The strata taken into account are the type of establishment and the territorial distribution by voivodeship. There were 23 shelters (7.8% of all shelters) and 18 hostels (16.2% of all hostels) drawn. The final analysis was performed on the group of 580 people, out of which 102 were women, and 478 were men, while 34 were excluded due to incompleteness of the data. The only exclusion criterion from the study was the lack of informed and voluntary consent of the subject. In the case of women, also pregnancy.

The examination of the homeless study subjects was approved by The Bioethics Committee of Jan Kochanowski University of Humanities and Sciences in Kielce (no. 05/2008 i No. 11/2012). The written informed consent was obtained from all study participants. All homeless people signed informed consent to participate in the study.

### 2.2. Anthropometric Measurements

Anthropometric measurements were performed using a stadiometer (GMP, Zurich, Switzerland) and a digital medical scale (Radwag WPT 100/200, Radom, Poland). In all measurements, height and body mass, the circumference of hip and waist were collected. Body mass index (BMI) was calculated according to the formula: mass in kg/ height m^2^, and classified according to the recommendations of the World Health Organization (WHO) as follows: underweight BMI < 18.5 kg/m^2^, normal body mass 18.5–24.9 kg/m^2^, overweight 25.0–29.9 kg/m^2^ and obesity 0 kg/m^2^ [17]. Abdominal obesity was calculated by measuring waist circumference in a vertical position, halfway between the lower costal margin and the upper margin of the iliac crest. Waist circumference >88 cm in women and >102 cm in men were determined abnormal, indicative of abdominal obesity.

### 2.3. Assessment of Biomarkers

As far as the selection of hematological parameters, these were as follows: red blood cells (RBC), hematocrit (HCT), mean corpuscular volume (MCV), hemoglobin (Hgb), mean corpuscular hemoglobin concentration (MCHC), mean corpuscular hemoglobin (MCH). For total lymphocyte count, i.e., immunological reserve, lymphocyte concentration was measured. The above-mentioned parameters were examined using analyzer Sysmex model XT 2000i, Sysmex model XS 1000i, and adequate original reagents from SYSMEX CORPORATION (distributor Sysmex Europe Gmbh Nordersted, Deutschland; Kobe, Japan) with normal values proposed by the manufacturer. There was DC detection (conductometric) used in the study, while hemoglobin concentration was measured using spectrophotometry, and analysis of white blood cell count was performed in flow cytometry. Furthermore, the following indicators were used: neutrophils-to-lymphocytes ratio (NLR), platelets-to-lymphocytes ratio (PLR), platelets-to-leukocytes ratio (PWR), the concentration of C-reactive protein (CRP). As for biochemical parameters, iron concentration, albumin and total protein concentration were analyzed in 386 individuals (81 women, 305 men). For measurements of the above-mentioned markers and total cholesterol, HDL, LDL, triglycerides and glucose fasting serum was used. Biochemical parameters were examined with the enzymatic method (spectrophotometry VIS) using the analyzer Cobas 6000/c501 (Tokio, Japan; distributor: Rosch Diagnostic, Warsaw, Poland) with reagents supported by the manufacturer. LDL concentration was calculated using the Friedwald formula (in people with TG < 400 mg%). All the concentrations of the markers were perfumed in a single certified laboratory. The blood for testing was collected in accordance with the laboratory instructions that describe in detail the collection and transport of the biological material that is the subject of the clinical trial.

The results were interpreted according to the norms provided by the European Cardiologic Association [18].

### 2.4. Statistical Analyses

Mean values, standard deviation, median and quartiles were used for the description of quantitative features. Quality features were described using frequency and percentages. The frequencies were compared with chi-squared or exact Fisher’s tests. Normal distribution was determined with Shapiro–Wilk test. If normal distribution of the data were found, Student’s *t*-test was chosen for comparison of distributions, while in opposite cases, U Mann’s-Whitney’s test was implemented. Raw and adjusted odds ratios (OR) with 95% of confidence interval 995% CI) were determined in the logistic regression model. All the statistical tests that were performed were two-sided. *p* < 0.05 was used for statistical significance. R software (version 3.6.0; R Core Team (2019). R: A language and environment for statistical computing. R Foundation for Statistical Computing, Vienna, Austria, URL https:/www.R-project.org/ (Statistical computing were performed in July 2000) was used for all the calculations.

## 3. Results

The majority of the individuals enrolled were single, with low-level education, mainly male, with smoking habits (Table 1). The results obtained were analyzed according to the gender within three age groups: 19–39, 40–59, 60–79 years. The dominant group was people aged 40–59 (54.3%), out of which 56.9% were female. The least numerous was the youngest group.

The chi-squared test results and the Fisher’s exact test allow (when the assumptions of the chi-squared test were not met) allow to state that the group of homeless people is significantly different in terms of marital status and smoking (Table 1).

### 3.1. Nutritional Status Based on Some Anthropometric Measurements

When analyzing nutritional status based on BMI, one may state that the mean values in both women and men did not differ significantly and reached values classified as overweight (Table 2).

Taking into consideration BMI classification in the analyzed group, the majority of the homeless people, i.e., 52.5%, were characterized by normal weight. Only a small percentage, i.e., 3.6%, were underweight. Overweight and obesity were found, respectively, in 29% and 14.8% of cases. Fisher’s exact test did not show any significant differences in BMI by gender. (Table 3).

Mean waist circumference in the analyzed group was 92.15 ± 13.77 and was significantly greater in men, 87.22 ± 15.30 cm vs.93.20 ± 13.20 cm, respectively (*p* < 0.0001). Abdominal obesity was found in 27.1% of cases, out of which 43.1% were women. As a consequence, a significantly larger group of the homeless (72.95%) women and men did not have abdominal obesity, i.e., 56.9% i 76.4%, respectively.

### 3.2. Nutritional Status Based on Results of Selected Complete Blond Count Parameters, Glucose, Lipid Profile, Iron, Albumin and Total Protein Levels and Immunological Reserve

After analysis of the selected elements of complete blood count, one can claim that in the majority of cases, homeless people were found to have normal values of the parameters. Most frequently, there were lower levels of red blood cells detected in men from the medium age group and hemoglobin concentration in the youngest women. Lower levels of hematocrit and leukocytes were detected in the oldest individuals, especially females. In terms of the analyzed indices, significant differences concern women and men in terms of hemoglobin concentration in two age groups (40–59 years and 60–79 years) and the RBC index in all age groups (Table 4).

Among analyzed markers in serum, there were abnormalities found in the mean concentration of total cholesterol and LDL cholesterol In the majority of men and women, they exceeded the normal values, respectively, in females: 64.4% and males: 67.6% vs. F: 61.6% and males: 68.8%. Elevated glucose level was detected in the oldest group, while the increased values of LDL cholesterol were found in women aged 40–59 years, and lowered HDL cholesterol in men and women from the oldest group. In the dominant percentage of the oldest women, there were abnormally elevated concentrations of triglycerides. Significant differences of the above markers in terms of gender concerned the concentration of total cholesterol and LDL cholesterol in the youngest age group (Table 4).

Iron concentration below normal range was found in 27% of women and in the minor group of men (8.5%). This phenomenon was reported in the oldest group of women. The lowered concentration of albumin was found only in 1% of individuals, and it was proved mainly in the oldest men. A greater percentage of men than women was diagnosed with elevated total protein, 11.1% vs. 6.2%, respectively. The above-mentioned abnormalities were detected mainly in the youngest man and in both genders in people aged 60–79. When analyzing the relationship between sex and the concentration of the above markers, significant differences were found in all age groups only in terms of iron concentration, while albumin in the youngest age group (Table 4).

When assessing the nutritional status based on mean values of CLL, it was found that in the studied group, the normal immunological reserve was observed 2676.30 ± 796.10 in 1 mm^3^, Me = 2618.70 (2099.46–3131.63) in 95.6% of cases without significant differences concerning men and women (*p* = 0.76). Abnormal values indicative of malnutrition were found in significantly less numerous groups (*p* < 0.05), i.e., 4.3% of individuals, more frequently in women than men (6.9% vs. 3.8%, respectively). The result of CLL within range 1200–1499 in 1 mm^3^ classified as minor malnutrition was detected in 3.3% of cases (3% of women and 3.4% of men). Moderate malnutrition was observed in only 1% of cases (women 4.0% vs. men 0.4%). Only in the oldest age group was there a significant difference in the immune reserve in relation to gender (Table 5).

Adjusted to age, gender, and smoking, the presence of abdominal obesity and concentration of analyzed blood markers parameters lead to the following observations (*p* < 0.05). The presence of abdominal obesity increased the risk of abnormalities in HCT, RBC, glucose level, total cholesterol level, HDL cholesterol level, triglycerides level, total protein level and complete lymphocyte count. In obese people, there was an increased risk of both increased and lowered HCT, lowered RBC, increased glucose level, total cholesterol, triglycerides and total protein (*p* < 0.05) (Table 6). Analysis of selected blood markers connected with nutritional status adjusted to gender, age, smoking and BMI revealed that BMI indicative of obesity significantly increases the risk of an increase in HCT and abnormal concentration of HDL cholesterol (*p* < 0.05). Overweight and obese people significantly more frequently were found to have a lower number of RBC, elevated glucose and triglycerides (Table 7).

### 3.3. Nutritional Status and Selected Indicators of Inflammatory Process

When analyzing selected indicators of the inflammatory process, one can state that the highest values of NLR were found in the oldest group. The mean value of PLR was highest in the oldest women and youngest men. PWR values were highest in women aged 60–79 years. Analysis of CRP concentration in the homeless people one can conclude that both in women and men, mean values of the marker were abnormally elevated. The highest CRP concentration was found in the oldest men and women (Table 8).

The values of novel biomarkers of inflammatory processes in women and men were analyzed according to BMI classification. It was found that significant differences were found only as far as PLR and PWR were concerned. In overweight and obese people, PLR values were significantly lower when compared with a group of people with normal BMI or underweight. Together with the increase of BMI, there was a significant decrease in mean values of PWR noted (Table 9).

## 4. Discussion

The recent study aimed at assessing the nutritional status of homeless adults in Poland. In the majority of individuals included in the analysis BMI—based weight was within limits of the norm (51% and 52.9% in females and males, respectively), with a low prevalence of underweight (2% and 4% females and males, respectively). Simultaneously overweight (F: 27.5%, M: 29.3%), as well as obesity (F: 19.6%, M: 13.8%) remained a common issue in the analyzed group. Due to the necessity to collect blood for testing and taking the necessary measurements, the test was limited to the homeless, who were temporarily staying in shelters or hostels in Poland. Due to the fact that it is difficult to clearly determine the number of homeless people, the list of institutions in Poland obtained from the Ministry of Labor and Social Policy was used during the draw. According to the above data, the number of places in hostels was 12,456 and in hostels 3632. As of the day of the study, the authors had no information on whether all of the above places were occupied by homeless people. When adjusted for age overweight rate in 2013–2014 was 30.5 and 43.2% in females and males, respectively, whereas obesity affected 25% and 24.4% females and males, respectively [19]. Analysis limited to single-province data (Wroclaw and surrounding areas) from 2018 yielded similar results—a total of 31% of participants were reported with obesity with no significant difference between males and females, while 36.7% of females and 48.1% of males were overweighted [20]. When compared to the general population of Polish adults, overweight and obesity rates in the homeless were noticeably lower. Underweight in the general adult population is yet to be reported in Poland.

There is constantly growing evidence for the lack of differences between the nutritional status of the homeless and non-homeless. A recent analysis from the USA reveals a similar BMI distribution pattern in the homeless and general population. Obesity was observed in up to 30% of analyzed homeless, whereas underweight was reported only in 1.6% of participants [21]. Another comprehensive analysis of the nutritional status of the homeless men from Rhode Island reported 39% obesity and 29.4% overweight rate [22]. The following tendency may be attributed to particular dietary habits (cheap, high in calories, periodically available and highly processed food). This remains consistent with previous observations suggesting that the homeless satisfy caloric demand with a higher intake of fats, compensating for a lower intake of proteins and hydrocarbons. It has been previously reported that intake of salt, meat, alcoholic beverages, potatoes is higher in the homeless when compared to non-homeless, whereas intake of fruits, vegetables and nuts is lower [23]. Considering the following, it can be assumed that overweight has become a major health-concern, whereas underweight may be considered less significant. When combined with smoking (F: 75.5%, M: 80.1%) and hypercholesterolemia (F: 63.4%, M: 67.6%) and hypertriglyceridemia (F26.7%, M25.8%), excessive body mass may significantly increase cardiovascular risk.

When seeking for reasons of overweight in the homeless, one should notice the previously reported correlation between sleep time and risk of higher BMI. Homeless people sleep mostly > 10 h and present a low-level of physical activity. Simultaneously, although daily sleep longer than 9 h contributes to a higher risk of abdominal (visceral) obesity, it has not been linked directly with HDL cholesterol and triglycerides serum levels [24,25,26].

The role of nutrition education in homeless shelters is well described in the literature. Both homeless and staff of canteens should be educated. What limits positive changes in food quality is insufficient funding and food donations. Meals served in the homeless shelters are rich in fat and sugar, whereas vegetable portions are minimal [27]. Although underweight remains uncommon in the homeless, the problem of quantitative malnutrition should not be ignored. It is suspected to result from not only an insufficient supply of calories but also from a deficit of micro- and macroelements, alcohol abuse and chronic conditions [28]. In the homeless alcoholics, a significant portion of energetic demand is satisfied with alcohol, which increases the risk of vitamins and mineral insufficiency and malnourishment. Since alcohol may damage the intestine mucosa, it can additionally affect nutrients absorption.

In this study, albumins, total protein, full blood count (WBC, hemoglobin, hematocrit, RBC), total cholesterol, triglycerides, HDL-C, LDL-C, glucose and iron were utilized for assessment of nutritional status. From all listed above, albumin, hemoglobin, total protein and total cholesterol are considered the most accurate malnutrition markers. It should be noticed that albumin levels may be inflammation-dependent, which is not an issue for BMI, total protein and hemoglobin. The prognostic value of WBC, hematocrit and triglycerides is questionable when predicting malnutrition [29]. Although the correlation of albumin levels with malnutrition provides a rationale for its clinical implementation, it is troublesome to define the cutoff value, especially in elderlies. Setting the threshold at the level of 3.5 g/dL may result in underdiagnosing elderlies with malnutrition [29,30]. In our study, hypoalbuminemia was observed in 1% of males and was not observed in females. Since malnutrition prevalence was also generally low (F: 2%, M: 4%), validation of albumin utility was not feasible. When considering further validation of albumin as a biomarker of nutritional status, utilizing a population with expected higher rates of malnutrition would be advocated.

Serum iron was decreased in 27.2% of females with iron deficiency affecting, in particular, women aged 60–79 (33%) and 8.5% of males. Since iron deficiency may result from several chronic conditions, including oncological diseases [31], its association with malnutrition remains uncertain. Moreover, the predominant prevalence of iron deficiency in the female population suggests gynecological conditions presenting with menorrhagia as potential confounders.

The present study has several limitations that should be signalized. Due to the low prevalence of malnutrition constituting the major end-point of the study (2% females and 4% males), analysis validating the utility of potential biomarkers failed to achieve statistical power. Due to missing data describing concomitant chronic conditions, it was not possible to rule out all potential confounders in the adjusted analysis.

What constitutes strong points of the study are the sample size and random collection of participants, which contributed to developing a representative sample. To the best of our knowledge, this is the first study evaluating the nutritional status of the homeless populations utilizing a consecutive sample and a complete list of clinically utilizable biomarkers of malnutrition.

The assessment of the nutritional status is to be the basis for improving the diet of the homeless people in welfare institutions and indirectly for improving their health. It is also the basis for the implementation of proper nutritional education and thus for changing health attitudes consisting in reducing risk behaviors, increasing awareness in the field of preventive healthcare, and stimulating motivation to care for health. Nutritional education, which is an important element of health education, implemented in relation to groups at risk of social exclusion, is considered one of the most important challenges for public health. A large part of it is part of the so-called NVAE, which is one of the pillars of the lifelong learning model.

## 5. Conclusions

Obesity and overweight in the homeless constitute major problems, whereas underweight is of marginal significance. The high prevalence of disorders included in metabolic syndrome coexisting with smoking habits indicate particular cardiovascular risk in the homeless population. According to BMI and biochemical markers analysis, malnutrition remains uncommon in the homeless, with mild malnutrition present in 3.3% and medium malnutrition present in 1%. Iron deficiency and anemia rates varied depending on sex, which may be attributed to particular pathologies of the female reproductive and hormonal system.

## Figures and Tables

**Table 1 ijerph-18-02340-t001:** Sociodemographic characteristics of the homeless.

Variable	Gender	Homeless Subjects			
		N	Prevalence (%)	95% CI	*p*
Marital status	F:	102	17.6	14.6–21.0	22.3, *p* < 0.0001 *
	M:	478	82.4	79.0–85.4	
Married	F:	17	16.7	10.3–25.6	
	M:	43	9.0	6.7–12.0	
Widowed	F:	19	18.6	11.9–27.8	
	M:	38	7.9	5.8–10.8	
Single	F:	19	18.6	11.9–27.8	
	M:	164	34.3	30.1–38.8	
Divorced/separated	F:	42	41.2	31.7–51.4	
	M:	228	47.7	43.2–52.3	
No data available	F:	5	4.9	1.8–11.6	
	M:	5	1.0	0.4–2.6	
Education	F:	102	17.6	14.6–21.0	*p* = 0.005 **
	M:	478	82.4	79.0–85.4	
Incomplete basic and basic	F:	44	43.1	33.5–53.3	
	M:	159	33.3	29.1–37.7	
Technical & incomplete secondary	F:	30	29.4	21.0–39.4	
	M:	231	48.3	43.8–52.9	
Secondary and incomplete higher	F:	22	21.6	14.3–31.0	
	M:	71	14.9	11.9–18.4	
Higher	F:	1	1.0	0.1–6.1	
	M:	12	2.5	1.4–4.5	
No data available	F:	5	4.9	1.8–11.6	
	M:	5	1.0	0.4–2.6	
Smoking status	F:	102	17.6	14.6–21.0	10.7, *p* = 0.0048 *
	M:	478	82.4	79.0–85.4	
Active smoker	F:	77	75.5	65.8–83.2	
	M:	383	80.1	76.2–83.6	
Smoking in the past	F:	8	7.8	3.7–15.3	
	M:	61	12.8	10.0–16.2	
Never smoking	F:	17	16.7	10.3–25.6	
	M:	34	7.1	5.0–9.9	

N—sample size, 95% CI—95% confidence intervals of prevalence; F—female; M—male; *p*—statistical significance; * chi-squared test, ** Fisher’s exact test.

**Table 2 ijerph-18-02340-t002:** BMI according to gender in the homeless people.

Variable	Gender	N	X ± SD
Body mass index (BMI)	M:	478	24.9 ± 4.8
	F:	102	25.7 ± 6.1
	Total	580	25.1 ± 5.0

N—sample size, X ± SD—mean ± standard deviation.

**Table 3 ijerph-18-02340-t003:** BMI in the homeless people according to gender in WHO classification.

Variable Body Mass Index	Gender	Homeless Subjects			
		N	Prevalence (%)	95% CI	*p*
Underweight, BMI < 18.5 kg/m^2^	F:	2	2.0	0.3–7.6	*p* = 0.430 *
	M:	19	4.0	2.5–6.2	
Normal weight, BMI = 18.5–24.9 kg/m^2^	F:	52	51.0	40.9–60.9	
	M:	253	52.9	48.3–57.5	
Overweight, BMI = 25.0–29.9 kg/m^2^	F:	28	27.5	19.3–37.3	
	M:	140	29.3	25.3–33.6	
Obese, BMI ≥ 30 kg/m^2^	F:	20	19.6	12.7–28.9	
	M:	66	13.8	10.9–17.3	

N—sample size; 95% CI—95% confidence intervals of prevalence; BMI categories as defined by WHO. *p*—statistical significance; * Fisher’s exact test.

**Table 4 ijerph-18-02340-t004:** Concentration of selected blood markers, glucose, total cholesterol, triglycerides, LDL-cholesterol, HDL-cholesterol, iron, albumin, total protein in the homeless people according to gender and age.

Variable	Age	Gender	N	X ± SD	Above Normal Range (%)	Normal Range (%)	Below Normal Range (%)	*p*
HGB Normal values: F: 12–16 g/dL M: 12–16.8 g/dL	18–39	F:	22	13.4 ± 1.08	13.6	86.4	0	*p* = 0.1287 **
		M:	57	14.87 ± 1.12	3.5	96.5	0	
	40–59	F:	58	13.43 ± 1.14	8.6	89.7	1.7	*p* = 0.0069 **
		M:	255	14.39 ± 1.04	2	98	0	
	60–79	F:	21	13.16 ± 1.74	19	76.2	4.8	*p* = 0.0019 **
		M:	161	14.29 ± 1.18	3.7	96.3	0	
	Total	F:	101	13.37 ± 1.26	11.9	86.1	2	*p* < 0.0001 **
		M:	473	14.41 ± 1.11	2.7	97.3	0	
HCT Normal values: F: 37.0–47.0%, M: 40.0–49.5%	18–39	F:	22	42 ± 2.92	0	100	0	*p* = 0.159 **
		M:	57	45.94 ± 3.32	5.3	84.2	10.5	
	40–59	F:	58	43.56 ± 4.08	3.4	77.6	19	*p* = 0.3622 **
		M:	255	45.33 ± 3.57	4.7	83.1	12.2	
	60–79	F:	21	42.92 ± 4.89	9.5	66.7	23.8	*p* = 0.4314 **
		M:	161	45.59 ± 3.99	8.1	77.6	14.3	
	Total	F:	101	43.09 ± 4.05	4	80.2	15.8	1.21, *p* = 0.5450 *
		M:	473	45.49 ± 3.69	5.9	81.4	12.7	
RBC Normal values: F: 3.50–5.00 T/L, M: 4.50–5.70 T/L	18–39	F:	22	4.41 ± 0.35	0	95.5	4.5	*p* = 0.005 **
		M:	57	4.82 ± 0.43	22.8	77.2	0	
	40–59	F:	58	4.45 ± 0.42	1.7	91.4	6.9	*p* < 0.0001 **
		M:	255	4.62 ± 0.40	37.6	62	0.4	
	60–79	F:	21	4.37 ± 0.47	4.8	81	14.3	*p* = 0.0015 **
		M:	161	4.68 ± 0.46	34.2	63.4	2.5	
	Total	F:	101	4.43 ± 0.42	2	90.1	7.9	*p* < 0.0001 **
		M:	473	4.66 ± 0.43	34.7	64.3	1.1	
WBC Normal values: F and M: 3.50–10.00 G/L	18–39	F:	22	7.54 ± 2.11	0	86.4	13.6	*p* = 0.1287 **
		M:	57	7.51 ± 1.49	0	96.5	3.5	
	40–59	F:	58	7.69 ± 2.36	0	87.9	12.1	*p* = 0.1702 **
		M:	255	8.33 ± 2.24	0.4	76.9	22.7	
	60–79	F:	21	7.70 ± 2.35	4.8	85.7	9.5	*p* = 0.0866 **
		M:	161	8.33 ± 1.93	0	814	18.6	
	Total	F:	101	7.66 ± 2.28	1	87.1	11.9	*p* = 0.0845 **
		M:	473	8.23 ± 2.08	0.2	80..8	19	
Glucose F and M: 60.0–99.9 mg/dL	18–39	F:	22	85.8 ± 9.44	0	86.4	13.6	*p* = 0.6652 **
		M:	57	96.5 ± 50.33	1.8	77.2	21.1	
	40–59	F:	58	100.4 ± 39.07	1.7	69	29.3	*p* = 0.1521 **
		M:	254	102.1 ± 52.93	0	65.4	34.6	
	60–79	F:	21	109 ± 45.5	0	61.9	38.1	*p* = 1 **
		M:	161	102.7 ± 44.02	0.6	63.4	36	
	Total	F:	101	99 ± 36.92	1	71.3	27.7	*p* = 0.2777 **
		M:	472	101.6 ± 49.68	0.4	66.1	33.5	
Total cholesterol F and M: <190 mg/dL	18–39	F:	22	169.2 ± 21.76	0	86.4	13.6	10.75, *p* = 0.0010 *
		M:	57	200 ± 54.33	0	45.6	54.4	
	40–59	F:	58	228.5 ± 40.71	0	22.4	77.6	1.31, *p* = 0.2532 *
		M:	254	213.1 ± 47.14	0	29.9	70.1	
	60–79	F:	21	225.9 ± 55.61	0	23.8	76.2	0.54, *p* = 0.4625 *
		M:	161	209.1 ± 43.62	0	31.7	68.3	
	Total	F:	101	215 ± 47.48	0	36.6	63.4	0.67, *p* = 0.4138 *
		M:	472	210.1 ± 47	0	32.4	67.6	
LDL cholesterol F and M: <115 mg/dL	18–39	F:	22	94 ± 20.11	0	86.4	13.6	9.08, *p* = 0.0026 *
		M:	55	120.4 ± 35.95	0	49.1	50.9	
	40–59	F:	57	143.4 ± 36.37	0	24.6	75.4	0.18, *p* = 0.6690 *
		M:	245	137.4 ± 39.8	0	27.3	72.7	
	60–79	F:	20	137.1 ± 34.44	0	25	75	0.30, *p* = 0.5816 *
		M:	158	133.8 ± 38.6	0	31	69	
	Total	F:	99	131.2 ± 38.43	0	38.4	61.6	1.90, *p* = 0.1677 *
		M:	458	134.2 ± 39.23	0	31.2	68.8	
HDL cholesterol F: >46 mg/dL M: >40 mg/dL	18–39	F:	22	56.4 ± 12.08	18.2	81.8	0	*p* = 0.767 **
		M:	57	51 ± 14.65	22.8	77.2	0	
	40–59	F:	58	58 ± 15.47	20.7	79.3	0	1.69, *p* = 0.1941 *
		M:	254	47.7 ± 12.95	29.1	70.9	0	
	60–79	F:	21	54 ± 15.12	38.1	61.9	0	0.42, *p* = 0.5150 *
		M:	161	48.6 ± 14.14	31.1	68.9	0	
	Total	F:	101	56.8 ± 14.67	23.8	76.2	0	1.14, *p* = 0.2855 *
		M:	472	48.4 ± 13.59	29	71	0	
Triglycerides F and M: <150 mg/dL	18–39	F:	22	94.5 ± 41.45	0	86.4	13.6	*p* = 0.5348 **
		M:	57	126.1 ± 108.49	0	77.2	22.8	
	40–59	F:	58	134.4 ± 64.25	0	70.7	29.3	0.07, *p* = 0.7884 *
		M:	254	135.7 ± 87.02	0	72.4	27.6	
	60–79	F:	21	139.2 ± 66.41	0	66.7	33.3	0.82, *p* = 0.3663 *
		M:	161	128 ± 67.56	0	75.8	24.2	
	Total	F:	101	126.7 ± 62.38	0	73.3	26.7	0.03, *p* = 0.8540 *
		M:	472	131.9 ± 83.87	0	74.2	25.8	
Iron F and M: 49–167 ug	18–39	F:	16	79.69 ± 39.98	25	75	0	*p* = 0.0012 **
		M:	36	98.29 ± 43.11	5.6	52.8	41.7	
	40–59	F:	50	60.8 ± 25.14	26	74	0	26.06, *p* < 0.0001*
		M:	165	90.93 ± 39.14	7.9	62.4	29.7	
	60–79	F:	15	70.73 ± 39.32	33.3	66.7	0	*p* = 0.0195**
		M:	104	79.11 ± 33.41	10.6	71.2	18.3	
	Total	F:	81	66.37 ± 31.87	27.2	72.8	0	40.63, *p* < 0.0001*
		M:	305	87.76 ± 38.24	8.5	64.3	27.2	
Albumin F and M: 3.5–5 g/dL	18–39	F:	16	4.33 ± 0.22	0	100	0	7.69, *p* = 0.0055*
		M:	36	4.51 ± 0.26	0	100	0	
	40–59	F:	50	4.24 ± 0.3	0	100	0	*p* = 1 **
		M:	165	4.35 ± 0.30	0.6	99.4	0	
	60–79	F:	15	4.13 ± 0.32	0	100	0	*p* = 1 **
		M:	104	4.23 ± 0.29	1.9	98.1	0	
	Total	F:	81	4.23 ± 0.29	0	100	0	*p* = 1 **
		M:	305	4.33 ± 0.31	1	99	0	
Total protein F and M: 6–8 g/dL	18–39	F:	16	7.27 ± 0.35	0	100	0	*p* = 0.1601 **
		M:	36	7.53 ± 0.49	0	83.3	16.7	
	40–59	F:	50	7.36 ± 0.45	0	94	6	*p* = 0.5737 **
		M:	165	7.48 ± 0.45	0	90.3	9.7	
	60–79	F:	15	7.39 ± 0.51	0	86.7	13.3	*p* = 0.6897 **
		M:	104	7.53 ± 0.51	0	88.5	11.5	
	Total	F:	81	7.35 ± 0.44	0	93.8	6.2	1.74, *p* = 0.1867 *
		M:	305	7.51 ± 0.48	0	88.9	11.1	

N—sample size; X ± SD—mean ± standard deviation; F—female; M—male; *p*—statistical significance; * chi-squared test, ** Fisher’s exact test.

**Table 5 ijerph-18-02340-t005:** Total leukocyte count in homeless people according to gender and age.

Variable	Age	Gender	N	X ± SD	Below Normal Range (%)	Normal Range (%)	*p*
CLL F and M: >1500/mm^3^	18–39	F:	22	2595 ± 798.26	4.5	95.5	*p* = 1 *
		M:	57	2570.2 ± 744.44	7	93	
	40–59	F:	58	2693.8 ± 799.5	3.4	96.6	*p* = 0.6176 *
		M:	255	2731 ± 770.73	2	98	
	60–79	F:	21	2618.7 ± 1038.6	19	81	*p* = 0.0472 *
		M:	161	2639.5 ± 821.38	5.6	94.4	
	Total	F:	101	2656.6 ± 846.11	6.9	93.1	*p* = 0.1779 *
		M:	473	2680.5 ± 785.84	3.8	96.2	

N—sample size; CLL—complete leukocyte level (immunological reserve); X ± SD—mean ± standard deviation; F—female; M—male; *p*—statistical significance * Fisher’s exact test.

**Table 6 ijerph-18-02340-t006:** Odds ratio for abnormal blood markers adjusted for age, gender, smoking.

Variable	N	Adjusted for Age and Gender		Adjusted for Age and Gender and Smoking	
		OR (95% CI)	*p*	OR (95% CI)	*p*
Hemoglobin below normal range	25	0.45 (0.14–1.19)	0.133	0.39 (0.12–1.07)	0.089
Hematocrit below normal range	32	0.25 0.06–0.73	0.025	0.26 (0.06–0.79)	0.035
Hematocrit above normal range	76	2.39 1.43–3.97	0.001	2.57 (1.51–4.33)	<0.001
Erythrocytes below normal range	166	0.55 0.34–0.87	0.013	0.57 (0.35–0.92)	0.024
Leukocytes above normal range	102	1.19 0.73–1.92	0.475	1.54 (0.93–2.54)	0.091
Glucose above normal range	186	2.38 1.6–3.53	<0.001	2.67 (1.77–4.03)	<0.001
Total cholesterol above normal range	383	1.54 1.01–2.37	0.046	1.59 (1.03–2.47)	0.039
LDL above normal range	376	1.51 0.98–2.37	0.067	1.54 (0.99–2.45)	0.061
HDL below normal range	161	2.75 1.83–4.12	<0.001	2.92 (1.92–4.45)	<0.001
Triglycerides above normal range	149	4.18 2.77–6.34	<0.001	4.06 (2.66–6.23)	<0.001
Iron below normal range	48	0.65 0.3–1.32	0.251	0.68 (0.3–1.41)	0.315
Iron above normal range	83	1.67 0.93–2.98	0.080	1.49 (0.81–2.69)	0.192
Total protein above normal range	39	3.15 1.57–6.33	0.001	2.66 (1.29–5.5)	0.008
Immunological reserve below normal range	25	0.74 0.26–1.83	0.533	0.55 (0.19–1.43)	0.249

N—sample size; OR—quotient of the odds of occurrence of a given event in the group of people with abdominal obesity compared to those without obesity; 95% Cl—confidence interval; *p*—statistical significance.

**Table 7 ijerph-18-02340-t007:** Odds ratio for abnormal blood markers adjusted for age, gender, smoking.

Variable	Comparison	Adjusted for Age and Gender		Adjusted for Age and Gender and Smoking	
		OR (95% CI)	*p*	OR (95% CI)	*p*
	Reference Values	1.00		1.00	
Hemoglobin below normal range	B vs. A	0.51 (0.18–1.41)	0.195	0.44 (0.15–1.27)	0.129
Hemoglobin below normal range	C vs. A	0.33 (0.07–1.51)	0.155	0.26 (0.05–1.24)	0.091
Hematocrit below normal range	B vs. A	0.45 (0.18–1.14)	0.092	0.48 (0.19–1.23)	0.128
Hematocrit below normal range	C vs. A	0.28 (0.06–1.21)	0.089	0.3 (0.07–1.37)	0.121
Hematocrit above normal range	B vs. A	1.41 (0.8–2.47)	0.231	1.49 (0.84–2.64)	0.167
Hematocrit above normal range	C vs. A	2.14 (1.13–4.04)	0.019	2.37 (1.22–4.58)	0.011
Erythrocytes below normal range	B vs. A	0.61 (0.39–0.94)	0.025	0.63 (0.40–0.99)	0.043
Erythrocytes below normal range	C vs. A	0.34 (0.17–0.65)	0.001	0.35 (0.18–0.68)	0.002
Erythrocytes above normal range	B vs. A	2.52 (0.65–9.78)	0.181	2.55 (0.64–10.09)	0.183
Erythrocytes above normal range	C vs. A	2.97 (0.69–12.74)	0.143	3.04 (0.65–14.12)	0.156
Leukocytes above normal range	B vs. A	0.66 (0.39–1.11)	0.119	0.79 (0.47–1.35)	0.394
Leukocytes above normal range	C vs. A	1.03 (0.56–1.89)	0.924	1.49 (0.79–2.82)	0.221
Glucose above normal range	B vs. A	1.49 (0.99–2.24)	0.054	1.65 (1.09–2.5)	0.019
Glucose above normal range	C vs. A	2.98 (1.81–4.90	<0.001	36 (2.12–6.1)	<0.001
Total cholesterol above normal range	B vs. A	1.34 (0.89–2.01)	0.163	1.41 (0.93–2.15)	0.104
Total cholesterol above normal range	C vs. A	1.56 (0.9–2.68)	0.111	1.67 (0.95–2.93)	0.076
LDL above normal range	B vs. A	1.34 (0.88–2.03)	0.169	1.39 (0.91–2.13)	0.130
LDL above normal range	C vs. A	1.61 (0.9–2.870)	0.105	1.7 (0.94–3.1)	0.081
HDL below normal range	B vs. A	1.21 (0.78–1.88)	0.387	1.3 (0.83–2.03)	0.246
HDL below normal range	C vs. A	4.35 (2.62–7.23)	<0.001	4.95 (2.89–8.46)	<0.001
Triglycerides above normal range	B vs. A	2.1 (1.35–3.28)	0.001	2.11 (1.34–3.32)	0.001
Triglycerides above normal range	C vs. A	6.26 (3.71–10.57)	<0.001	6.24 (3.63–10.74)	<0.001
Iron below normal range	B vs. A	0.62 (0.28–1.34)	0.224	0.63 (0.29–1.39)	0.256
Iron below normal range	C vs. A	0.73 (0.3–1.81)	0.498	0.77 (0.3–1.99)	0.595
Iron above normal range	B vs. A	1 (0.55–1.83)	0.999	0.85 (0.45–1.59)	0.613
Iron above normal range	C vs. A	1.89 (0.94–3.81)	0.076	1.58 (0.76–3.28)	0.223
Total protein above normal range	B vs. A	1.51 (0.67–3.38)	0.320	1.21 (0.52–2.8)	0.657
Total protein above normal range	C vs. A	3.27 (1.44–7.44)	0.005	2.5 (1.05–5.99)	0.039
Immunological reserve below normal range	B vs. A	1.35 (0.56–3.23)	0.501	1.02 (0.41–2.55)	0.959
Immunological reserve below normal range	C vs. A	0.81 (0.22–2.95)	0.752	0.52 (0.14–2.03)	0.350

A—normal weight + underweight, BMI < 24.9 kg/m^2^; B—overweight, BMI = 25.0–29.9 kg/m^2^; C—obesity, BMI ≥ 30.0 kg/m^2^ OR-odds ratio; 95% CI—confidence interval; *p*—statistical significance.

**Table 8 ijerph-18-02340-t008:** Mean values of neutrophils-to-lymphocytes ratio (NLR), platelets-to-lymphocytes ratio (PLR) and platelets-to-leukocytes ratio (PWR) and concentration of C-reactive protein (CRP) in the homeless people according to gender and age.

Variable	Age	Gender	N	X ± SD	Me (Q25–Q75)	Min–Max	*p*
NLR	18–39	F:	22	1.63 ± 0.51	1.51 (1.32–1.84)	0.79–2.63	0.012 A
		M:	57	1.66 ± 0.71	1.58 (1.23–1.81)	0.75–4.43	
		Total	79	1.65 ± 0.66	1.58 (1.24–1.83)	0.75–4.43	
	40–59	F:	58	1.57 ± 0.77	1.48 (1.00–1.89)	0.52–5.37	
		M:	255	1.75 ± 0.80	1.59 (1.23–2.03)	0.42–5.24	
		Total	313	1.71 ± 0.80	1.55 (1.19–2.01)	0.42–5.37	
	60–79	F:	21	1.87 ± 1.25	1.47 (1.09–1.94)	0.75–5.96	
		M:	161	1.93 ± 1.00	1.75 (1.26–2.39)	0.52–7.64	
		Total	182	1.92 ± 1.03	1.73 (1.25–2.35)	0.52–7.64	
	Total	F:	101	1.64 ± 0.85	1.5 (1.09–1.89)	0.52–5.96	0.105 G
		M:	473	1.80 ± 0.87	1.62 (1.24–2.09)	0.42–7.64	
PLR	18–39	F:	22	99.16 ± 59.16	81.52 (70.24–101.39)	58.90–340.59	0.355 A
		M:	57	109.25 ± 59.61	95.83 (73.43–121.47)	49.85–413.37	
		Total	79	106.44 ± 59.28	91.87 (72.45–118.03)	49.85–413.37	
	40–59	F:	58	106.27 ± 43.43	98 (77.97–122.04)	38.19–276.44	
		M:	255	97.23 ± 35.71	88.89 (74.19–112.78)	29.30–269.4	
		Total	313	98.90 ± 37.35	92.74 (74.74–115.39)	29.30–276.44	
	60–79	F:	21	113.72 ± 53.02	95.83 (71.69–145.58)	53.87–236.09	
		M:	161	101.10 ± 48.24	93.21 (66.23–125.53)	15.92–409.79	
		Total	182	102.56 ± 48.83	93.51 (67.14–126.56)	15.92–409.79	
	Total	F:	101	106.27 ± 48.94	95.4 (74.16–125.29)	38.19–340.59	0.201 G
		M:	473	99.99 ± 43.72	91.9 (71.86–118.36)	15.92–413.37	
PWR	18–39	F:	22	32.70 ± 13.55	29 (22.85–39.16)	19.28–81.9	0.031 A
		M:	57	34.71 ± 11.86	32.09 (28.37–38.5)	19.04–89.68	
		Total	79	34.15 ± 12.30	31.80 (26.31–39.09)	19.04–89.68	
	40–59	F:	58	36.52 ± 11.18	34.85 (29.43–40.81)	11.83–73.12	
		M:	255	31.21 ± 9.35	29.93 (24.41–37.46)	10.76–65.98	
		Total	313	32.19 ± 9.92	31.25 (25.00–38.62)	10.76–73.12	
	60–79	F:	21	35.76 ± 12.10	33.1 (28.35–43.85)	17.42–60.48	
		M:	161	29.86 ± 10.11	28.14 (23.62–33.77)	6.24–71.18	
		Total	182	30.19 ± 10.50	28.61 (23.75–34.35)	6.24–71.18	
	Total	F:	101	35.53 ± 11.89	33.93 (27.71–40.33)	11.83–81.9	<0.001 G
		M:	473	31.17 ± 10.03	29.73 (24.36–36.11)	6.24–89.68	
CRP	18–39	F:	22	1.88 ± 1.41	1.05 (1.00–2.13)	1.00–5.50	0.004 A
		M:	57	3.37 ± 6.64	1.10 (1.00–2.50)	1.00–38.60	
		Total	79	2.95 ± 5.72	1.10 (1.00–2.30)	1.00–38.60	
	40–59	F:	58	3.72 ± 5.63	1.55 (1.00–3.38)	1.00–31.00	
		M:	254	5.154 ± 7.69	2.20 (1.10–5.00)	1.00–53.50	
		Total	312	4.89 ± 7.36	2.10 (1.00–4.83)	1.00–170.00	
	60–79	F:	21	11.88 ± 31.48	2.00 (1.60–5.30)	1.00–144.90	
		M:	161	7.30 ± 16.59	2.80 (1.30–8.00)	1.00–170.50	
		Total	182	7.82 ± 18.84	2.65 (1.40–7.48)	1.00–170.50	
	Total	F:	101	5.02 ± 15.16	1.7 (1.00–3.3)	1.00–144.90	0.626 G
		M:	472	5.668 ± 11.4965	2.2 (1.1–5.025)	1–170.50	

N—sample size; NLR—neutrophil-to-lymphocyte ratio; PLR-platelet-to-lymphocyte ratio, PWR—platelet-to-white blood cell ratio; CRP—C reactive protein; X ± SD—mean ± standard deviation; Me—median (Q25–Q75—quartile 1–quartile 3); F—female; M—male; *p*—statistical significance: A—according to age, G—according to gender.

**Table 9 ijerph-18-02340-t009:** Mean values of NLR, PLR and PWR and concentration of CRP in homeless people according to BMI classification.

Variable	No Data	N	BMI Classification	X ± SD	Me (Q25–Q75)	*p*
NLR	4	326	A	1.83 ± 0.91	1.62 (1.23–2.12)	0.134
	1	168	B	1.66 ± 0.81	1.51 (1.14–2.03)	
	1	86	C	1.76 ± 0.80	1.58 (1.31–2.01)	
	6	580	Total	1.77 ± 0.87	1.60 (1.22–2.06)	
PLR	4	326	A	106.67 ± 49.38	98.81 (75.44–121.71)	0.003
	1	168	B	93.84 ± 36.94	88.64 (68.63–112.35)	
	1	86	C	94.24 ± 36.68	88.66 (69.28–115.92	
	6	580	Total	101.10 ± 44.70	92.98 (72.08–119.08)	
PWR	4	326	A	32.84 ± 10.58	30.73 (25.44–38.91)	0.043
	1	168	B	31.23 ± 10.97	29.93 (23.59–37.99)	
	1	86	C	29.92 ± 8.87	29.58 (23.57–33.99)	
	6	580	Total	31.94 ± 10.50	30.28 (24.63–37.85)	
CRP	5	326	A	6.13 ± 13.50	1.90 (1.00–4.90)	0.056
	1	168	B	3.70 ± 4.62	1.90 (1.00–4.25)	
	1	86	C	7.02 ± 16.39	3.30 (1.90–5.60)	
	7	580	Total	5.55 ± 12.21	2.10 (1.00–4.80)	

N—sample size; NLR—neutrophil-to-lymphocyte ratio; PLR-platelet-to-lymphocyte ratio; PWR—platelet-to-white blood cell ratio; CRP—C reactive protein; A—normal weight + underweight, BMI < 24.9 kg/m^2^; B—overweight, BMI = 25.0–29.9 kg/m^2^; C—obesity, BMI ≥ 30.0 kg/m^2^; X ± SD—mean ± standard deviation; Me—median (Q25–Q75—quartile l 1–quartile 3); *p*—statistical significance.

## Data Availability

The data presented in this study are available on request from the corresponding author.

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
