# Peer review of "Analysis of the Nutritional Status in Homeless People in Poland Based on the Selected Biochemical Parameters"

_ijerph, 2021, doi:10.3390/ijerph18052340_

Round 1

Reviewer 1 Report

Please see the attached. Thank you very much.

Reviewer 2 Report

I really appreciate the opportunity to review this manuscript entitled “Analysis of the nutritional status in homeless people in Poland based on the selected biochemical parameters”. This is important to assess the health status of homeless people.  I remark some issues (most of them in methods) in order to improve the quality of this manuscript.

The abstract is clear but it is important to explain the use of the abbreviations, for example about anthropometric indicators and biomarkers. Introduction was well structure. The aim is concise and I think it should be justify, why it is important to assess the nutritional status of homeless people?

At the methods section, there are some questions that should be review. Authors do not remark use of any kind of informant consent for the participants or to follow Helsinki Guidelines, and did they ask for the approval to an ethical committee? About the inclusion and exclusion criteria, do the authors take into account any? If they did, they should include them. Relate to the statistical analysis, why did they choose median and quartiles to describe quantitative measures?

Results were clear, but be careful in table 7 with the use of “ponizej” which is not English. Discussion summarize and explain in a good way the finding but, from my point of view there would be it would be interesting to discuss if it would be necessary or possible to assess another biomarker, for example Vitamin D. And if there is a great concern about nutritional education, what could be accomplish?

Conclusions were correct. References should be revise, underlined should be avoid. 

Round 2

Reviewer 1 Report

please see the attached.

Author Response

Dear Reviewer

Thank You for your time that You have spent on reviewing the article.

Authors

Reviewer 2 Report

Authors take into account the comments but they did not answer some questions in the coverletter about the justification and discussion.

Author Response

Dear Reviewer

Thank You for your time that You have spent on reviewing the article.

Please see the attached. We added parts in purple.

Authors
